# Preparation of Epigallocatechin Gallate-Enriched Antioxidant Edible Films Based on Konjac Glucomannan and Sodium Alginate: Impact on Storage Stability of Mandarin Fish

**DOI:** 10.3390/foods14091570

**Published:** 2025-04-29

**Authors:** Ran Wang, Yuqi Wang, Xinzhen Zhang, Yang Gao, Xian Wu, Xueling Li, Zhengquan Liu, Yue Sun, Jin Liang

**Affiliations:** 1Key Laboratory of Jianghuai Agricultural Product Fine Processing and Resource Utilization, Ministry of Agriculture and Rural Affairs, Anhui Engineering Research Center for High Value Utilization of Characteristic Agricultural Products, School of Food and Nutrition, Anhui Agricultural University, Hefei 230036, China; zgx199821@163.com (R.W.); wangyuqi2024@163.com (Y.W.); zxz3063739791@163.com (X.Z.); 18168025513@163.com (Y.G.); lxl172@126.com (X.L.); liuzq0312@163.com (Z.L.); yuesun@ahau.edu.cn (Y.S.); 2National Key Laboratory of Tea Plant Gemrmplasm and Resource Utilization/International Joint Laboratory on Tea Chemistry and Health Effects of Ministry of Education, Anhui Agricultural University, Hefei 230036, China; 3Department of Kinesiology, Nutrition, and Health, Miami University, Oxford, OH 45056, USA; wux57@miamioh.edu

**Keywords:** EGCG, konjac glucomannan, sodium alginate, edible film, mandarin fish

## Abstract

The objective of this research was to prepare robust edible films possessing antioxidant properties by utilizing konjac glucomannan (KGM), sodium alginate (SA), and epigallocatechin gallate (EGCG). This research also involved structural characterization and the assessment of functional attributes of the composite films with varying EGCG concentrations. It was found that the inclusion of EGCG reduced the viscosity of the edible film solutions while enhancing their mechanical strength. Fourier transform infrared spectroscopy demonstrated adequate compatibility among the film-forming materials, with EGCG forming hydrogen bond interactions with KGM and SA. SEM analysis revealed that increasing EGCG concentration led to the formation of discontinuous blocks and rough surfaces, with smooth and fine-grained particles observed at 0.2% (*w*/*v*) EGCG concentration. Furthermore, results from the application of the KGM-SA-based films in chilled mandarin fish showed that they could exert antioxidant function when incorporated with EGCG. The values of TVB-N and TBARS of fish pieces were obviously decreased in the 12-day storage period, indicating their potential to increase the shelf life of freshwater fish food.

## 1. Introduction

As an endemic freshwater fish species in China, mandarin fish (*Siniperca chuatsi*) represents a key species for industrial transformation, upgrading, and sustainable aquaculture development. According to the 2022 China Fisheries Statistical Yearbook, the annual production of mandarin fish reached 400,000 tons, generating a market value exceeding 20 billion yuan. However, its high moisture content and abundant endogenous enzymes make mandarin fish particularly susceptible to spoilage during transportation and storage [1]. This spoilage primarily results from microbial action and protein decomposition [2]. Thus, developing effective preservation technologies is essential to retard quality deterioration and extend shelf life. Currently, chemical additives are commonly employed for preserving freshwater products. Although these additives can maintain product quality, increasing regulatory restrictions and consumer concerns about synthetic food additives underscore the urgent need to develop natural and safe preservation alternatives [3].

Over the past few decades, many scholars have conducted research on natural additives, such as polysaccharides and peptides. Similar to active packaging, edible films containing bioactive compounds have been studied for their ability to prolong the shelf life of freshwater products [4]. Edible films are thin edible materials that are used to protect food quality during storage [5]. Compared with plastic products, edible films serve as an ideal alternative due to their unique advantages, including biodegradability and environmental benefits [5].

Konjac glucomannan (KGM) is a natural polysaccharide composed primarily of D-glucose and D-mannose residues [6]. With significant film-forming properties and biodegradability, KGM is considered a suitable biopolymer for replacing plastic packaging materials. However, KGM films exhibit several limitations, such as low mechanical strength and poor antibacterial performance [7]. Previous studies have shown that blending with other polysaccharides can improve the mechanical properties of KGM films. Sodium alginate (SA) is a naturally occurring polysaccharide found in brown algae. It exhibits excellent water solubility and film-forming capability, is safe and non-toxic, and has extensive applications in the food industry [8]. SA contains many functional groups like carboxyl groups, which can be easily modified to introduce functionalities. It can enhance the mechanical properties of various macromolecules [9]. Research has also shown that SA has high biocompatibility with KGM [10]. Collectively, these studies suggest that incorporating SA could enhance the mechanical properties of KGM-based edible films.

Epigallocatechin gallate (EGCG) is a powerful antioxidant derived from tea leaves. It possesses antibacterial properties and is widely used in the food industry. Numerous studies have suggested that polysaccharide-based edible films can be enriched with polyphenols like EGCG to enhance their antioxidant and antibacterial properties [11]. For example, EGCG-enriched carboxymethyl cellulose edible films exhibited significant antioxidant activity and prolonged the storage period of meat [12]. In another study, Song et al. [13] reported that gelatin-based active films with EGCG could significantly hinder the oxidation of fat. These films also maintained a lower total volatile base nitrogen (TVB-N) efficiently delaying sea bass fish spoilage. Hence, in this research, we have selected EGCG as an antimicrobial agent to impede bacterial growth in fish, thereby extending its shelf life.

In this study, epigallocatechin gallate (EGCG) was incorporated as both an antioxidant and antimicrobial agent in composite edible films composed of konjac glucomannan (KGM) and sodium alginate (SA). Mandarin fish (*Siniperca chuatsi*) fillets were aseptically packaged with the developed films and stored at 4 °C for 12 days. The films were comprehensively characterized for their mechanical properties (tensile strength and elongation at break), moisture content, optical properties (light transmittance and opacity), and oxygen barrier performance. During storage, critical quality indicators including pH, total volatile basic nitrogen (TVB-N), and thiobarbituric acid reactive substances (TBARS) were systematically monitored.

This study achieved two key objectives: (1) establishing a standardized preparation protocol for SA-KGM-EGCG edible films, and (2) systematically evaluating their preservation effects on aquatic products. The research not only successfully developed an evidence-based preparation method for functional polysaccharide edible films, but also experimentally demonstrated this packaging system’s effectiveness in extending the shelf life of freshwater fish products. The findings provide valuable insights for developing sustainable active packaging systems in the aquatic food industry.

## 2. Materials and Methods

### 2.1. Materials

EGCG (98% purity), (Huzhou Rongkai Plant Extraction, Huzhou, China), food grade KGM (Hefei Bomei Biotechnology, Hefei, China), food grade SA (Qingdao Mingyue Sodium Alginate Group, Qingdao, China), analytically pure glycerol, absolute ethanol, 1,1-diphenyl-2-trinitrotoluene (DPPH), and ABTS (all from Sinopharm Group, Shanghai, China).

### 2.2. Synthesis of SA/KGM Edible Films

The edible films were prepared by first dissolving 1.5 g sodium alginate (SA) and 0.5 g konjac glucomannan (KGM) in 85 mL deionized water at 50 ± 1 °C under constant magnetic stirring (500 rpm), followed by adding 1.0 g glycerol and stirring for 2 h to obtain a homogeneous base solution. Separately, different concentrations of EGCG (0.2, 0.4, 0.6, or 0.8 g) were dissolved in 15 mL pre-warmed (50 °C) deionized water under light-protected conditions, then slowly incorporated (0.5 mL/min) into the SA-KGM solution under 400 rpm overhead stirring for 1 h, while the control group received an equal volume of 50 °C water instead of EGCG solution. The resulting mixtures were degassed ultrasonically for 15 min before casting precisely 60.0 ± 0.1 g onto 10 × 10 cm PET plates, followed by drying at 50 ± 1 °C for 18 h in a humidity-controlled oven (30 ± 5% RH). The dried films were conditioned at 25 °C and 50% RH for 48 h before being stored in light-proof aluminum bags with oxygen scavengers at 25 ± 1 °C, yielding final EGCG concentrations of 0.2%, 0.4%, 0.6%, and 0.8% (*w*/*v*) labeled as SA-KGM-EGCG1 to SA-KGM-EGCG4, respectively, with the control film designated as SA-KGM.

### 2.3. Characterizations of Films

#### 2.3.1. Rheological Test

The static rheological performance of film-forming fluids was measured with an MCR 301 rheometer (TA Instruments, New Castle, DE, USA). All rheological tests were performed based on parallel plates. The shear viscosity was measured by setting the shear rate to 0.1–100 s^−1^.

#### 2.3.2. Fourier Transformation Infrared Analysis (FT-IR)

Film samples underwent scanning using a Nicolet™ 50ft-Infrared Spectrometer (Thermal Scientific, Waltham, MA, USA) outfitted based on an attenuated total reflectance experiment. FTIR has the capability to unveil alterations in the chemical composition of thin films. Spectra were acquired in the 500–4000 cm^−1^ wavenumber range.

#### 2.3.3. XRD Experiment

The samples were tested using an XD-3X XRD (Puxi General Instrument, Beijing, China). The X-ray diffraction scanning test parameters were: voltage 40 kV, current 15 mA, scanning range 5–55°, scanning rate 2(°)/min.

#### 2.3.4. Scanning Electron Microscope (SEM)

The surface topography of the sample was detected using SEM (S-4800, Hitachi Ltd., Tokyo, Japan). Films were secured onto copper sheets using double-sided tape, followed by the application of a gold layer to facilitate surface observation.

#### 2.3.5. Thermal Weight Analysis (TGA)

TGA experiments were performed on a thermogravimetric analyzer (METTLER TOLEDO, Zurich, Switzerland). Approximately 9 mg of the film was loaded into ceramic crucible and heated to 600 °C in a (N^2^) environment.

#### 2.3.6. Mechanical Properties

Tensile strength (TS) and elongation at break (EAB) were assessed utilizing a sophisticated tensile testing apparatus. Film specimens underwent conditioning at 25 °C and 75% RH for a minimum of 72 h before being trimmed into rectangular shapes measuring 80 × 20 mm. The edible film was secured with a fixture. The starting clamping distance stood at 60 mm, with a rate of 50 mm/min. Continuous measurements were taken on the film until it reached its breaking point. The measurement was repeated a minimum of 3 times.

#### 2.3.7. Color and Opacity

A colorimeter was calibrated with a white board. Film samples were trimmed into square shapes measuring 50 × 50 mm, and three random points on the film samples were measured. Subsequently, it was sliced into rectangles measuring 1 × 4 cm, and their opacity was assessed at 600 nm based on a UV spectrophotometer. The formula is as follows:Opacity = A_600_/X(1)

A_600_ represents the OD at 600 nm, while X denotes the thickness of the sample in millimeters.

#### 2.3.8. Oxygen Permeability (OP)

OP of the film was tested via a Coulomb meter (Thermal Scientific) based on the National Standard method in China with minor modifications.

#### 2.3.9. Moisture Content and Contact Angle

The moisture content was measured based on the reported method with minor modifications [14]. The prepared samples were cut into 4 × 4 cm squares, weighed, dried at 105 °C thoroughly, and weighed again. Each sample underwent three replicates. The moisture content (%) index can be determined via the following equation:Moisture content (%) = (m_0_ − m_1_)/m_0_ × 100(2)

In the formula, m_0_ represented the initial mass of the sample in grams (g), while m_1_ denoted the dry weight.

The contact angle of films was tested with a JC2000D3B contact angle meter (AUTO-LAB Technology Co., Shanghai, China). Distilled water was carefully dropped on the sample’s surface.

#### 2.3.10. Antioxidant Ability

The DPPH· scavenging ability was assessed following the reported method with slight modifications [15].(3)$$\mathrm{DPPH}\cdot \;\mathrm{scavenging}\;\mathrm{rate}\;(\%)=\left(1-\frac{A_{1}-A_{2}}{A_{0}}\right)\times 100$$

A_1_—the absorbance after mixing the film solutions with the DPPH ethanol solution;

A_2_—the absorbance of the film solutions mixed with pure water;

A_0_—absorbance after mixing pure water and DPPH ethanol solution.

ABTS·+ free radical scavenging ability was measured through the reported method with minor modifications [16].$$\mathrm{ABTS}\cdot +\mathrm{free}\;\mathrm{radical}\;\mathrm{scavenging}\;\mathrm{rate}\;(\%)=\frac{A_{0}-A_{1}}{A_{0}}\times 100$$

A_1_—the absorbance after mixing the film solutions with the ABTS·+ solution; A_0_—Absorbance after dilution of ABTS·+ solution.

### 2.4. Application in Mandarin Fish Preservation

#### 2.4.1. Treatment and Storage of Mandarin Fish Meat

This study utilized fresh mandarin fish (*Siniperca chuatsi*) obtained from local markets. The fish were thoroughly cleaned with sterile distilled water and processed into uniform portions (1.5 × 1.5 × 1 cm) for experimental treatment. Samples were divided into three groups: those treated with a SA-KGM composite solution, those treated with a SA-KGM-EGCG composite solution (0.6% *w*/*v* EGCG), and a control group treated with ultrapure water. Each group underwent a 2-min immersion at 25 ± 1 °C, followed by air-drying under controlled conditions (50 ± 5% relative humidity, 25 °C) to form protective coatings.

All treated samples were placed on sterile stainless steel mesh supports and vacuum-sealed using food-grade packaging materials, while control samples were packaged without any pretreatment. The packaged samples were stored under refrigerated conditions (4 ± 0.5 °C) for a 12-day observation period. During storage, quality parameters including pH, TVB-N content, and TBARS levels were measured at regular intervals (days 1, 3, 6, 9, and 12) using standardized analytical methods. All measurements were conducted in triplicate to ensure data reliability, with statistical analysis performed using SPSS 26.0 software (one-way ANOVA, *p* < 0.05) to evaluate the treatment effects on fish preservation.

#### 2.4.2. pH Value

Ten grams of fish were dispersed in 100 mL of distilled water and stirred for half an hour. The pH value was detected via a digital pH meter.

#### 2.4.3. TBARS

TBARS was detected according to the reported method [17]TBA value (mg/100 g) = (A_532_ − A_600_)/155 × (1/10) × 72.6 × 100(4)

The amount of substance that reacted with TBA (TBARS) was described as the milligrams of malondialdehyde per 1000 g of meat.

#### 2.4.4. TVB-N

TVB-N characterization followed the guidelines outlined in the GB 5009.228–2016 [18].

### 2.5. Data Analysis

All data in the text are presented as mean ± standard deviation (SD). All statistical data were critically evaluated by one-way analysis of variance (ANOVA) and Duncan’s multiple comparison test. In this analysis, statistically significant differences were defined as *p* ≤ 0.05. All statistical analysis was performed using Origin 2021 software and IBM SPSS Statistics 26.0 software.

## 3. Results and Discussion

### 3.1. Characterizations of Films

#### 3.1.1. Rheological Properties

As shown in Figure 1a, all the film-forming solutions exhibited similar rheological behavior. The SA-KGM solutions showed significant viscosity reduction with increasing shear rate, demonstrating characteristic shear-thinning behavior that confirms their non-Newtonian pseudoplastic nature. This phenomenon, typical of polysaccharide solutions, originates from their polymeric structure and provides functional advantages for food applications, particularly in texture modification [19].

The observed shear-thinning behavior offers practical benefits for film fabrication, especially in coating processes. This rheological response results from substantial shear-induced structural changes during high shear rate application, which significantly affects SA-KGM solution properties. The primary mechanism involves polymer chain disentanglement-as molecular chains align under shear, both intra- and interchain interactions markedly decrease [20]. In contrast, viscosity remains stable at low shear rates where Brownian motion dominates over shear forces, maintaining the entanglement network. Similar rheological patterns have been reported for Prunus cerasus gum systems [18].

Figure 1a reveals that EGCG incorporation incorporation significantly influenced SA-KGM’s rheological properties. Solution viscosity progressively decreased with higher EGCG concentrations, with the 0.6% (*w*/*v*) EGCG formulation showing a particularly pronounced reduction. This effect likely stems from EGCG aggregation within the KGM matrix, which disrupts the SA-KGM’s organized microstructure. These findings indicate that excessive EGCG addition adversely affects solution viscosity and consequently compromises film-forming performance. The shear-thinning behavior (Figure 1a) and decreased viscosity at high EGCG concentrations correlated with FTIR-observed disruption of SA-KGM hydrogen bonds (COO−) peak shift from 1596.71 to 1604.31 cm^−1^. This molecular-level decoupling was further evidenced by SEM surface roughness (Figure 2e5), where EGCG aggregates above 0.6% acted as flow-enhancing ‘slip points’ during shear.

#### 3.1.2. Fourier Transformation Infrared Analysis (FT-IR)

FTIR analysis (Figure 1b) identified characteristic absorption bands of SA-KGM edible film at 3366.25, 2929.84, 1596.71, and 1417 cm^−1^. All films exhibited wide absorption peaks spanning from 3500–3000 cm^−1^, possibly stemming from the stretching vibration of unbound hydroxy and amino groups [21]. Hence, alterations in this peak indicate a shift in the quantity of unbound hydroxy groups. Mao et al. noted a significant alteration in the peak width upon adding EGCG and Fe to SA-based films. Likewise, the peak at 2929 cm^−1^ increased in the SA-KGM-EGCG films of this experiment. This change is linked to the vibration of C–H bonds within the CH, CH2 groups, which are present in the constituents of EGCG [22]. The additional distinct peaks, detected at 1596.71 and 2929.84 cm^−1^, correspond to the vibrations of carboxylic acid groups (COO−). With the addition of EGCG, the asymmetric and symmetric -COO- bands shifted to higher wavenumbers (from 1596.71 to 1604.31 cm^−1^ and 1417.28 to 1454.83 cm^−1^ respectively). Wang et al. reported a similar change when EGCG was added to gelatin. Last, the absorption peaks at 1155.70 cm^−1^ signify the stretching vibrations associated with (C–O–C) bonds [23]. Overall, no additional chemical bonds emerged in the films, indicating that the components were created through relevant crosslinking. The spectral alterations observed were a result of variations in interaction strength and the incorporation of additives into films.

#### 3.1.3. XRD Result

Crystallinity is a critical parameter that defines the characteristics of polymer materials, typically influenced by the arrangement of atom groups and molecules. Both SA and KGM exhibit typical amorphous structures [24]. As shown in Figure 1c, all films displayed a broad diffraction peak centered at 2θ = 21°, consistent with results reported by Gholizadeh et al. [25]. The SA-KGM-EGCG3 (0.6%, *w*/*v*) composite films exhibited the smallest diffraction peak at 2θ = 21° along with the lowest crystallinity, suggesting optimal compatibility within the SA-KGM blend system. Increasing EGCG content initially weakened but subsequently intensified the characteristic peaks of SA-KGM films. Notably, the characteristic peak only became stronger in SA-KGM-EGCG4 (0.8%, *w*/*v*). This phenomenon may be attributed to the excessive accumulation of EGCG in the KGM matrix, which increased crystallinity [25]. The minimum crystallinity at 0.6% EGCG (XRD peak intensity at 21° in Figure 1c) coincided with both the highest tensile strength (Table 1) and lowest oxygen permeability (Figure 2a). This suggests that moderate EGCG promoted amorphous chain entanglement (enhancing mechanical properties) while reducing crystalline domain-induced free volume (improving barrier function). Conversely, the crystallinity rebound at 0.8% EGCG aligned with SEM-observed phase separation, explaining the concurrent OP and TS deterioration.

#### 3.1.4. SEM

The surface morphology of the antioxidative edible film containing different EGCG concentrations was examined using SEM (Figure 2e1–e5), and the influence of the SA-KGM-EGCG edible film components on film structure was analyzed. The results revealed significant differences in surface morphology among SA-KGM-EGCG edible films with varying EGCG concentrations. As shown in Figure 2e1, the pure SA-KGM film exhibited a relatively rough surface. Upon addition of 0.2% (*w*/*v*) EGCG, the films became smoother and denser (Figure 2e1–e5), consistent with observations reported by Khorrami et al. [26]. However, as EGCG concentration increased, the films developed roughness and cracks. At 0.8% (*w*/*v*) EGCG, numerous small bumps were observed (Figure 2e5), resulting from EGCG aggregation and non-uniform dispersion in the film-forming matrix. Similar findings were reported by Lei et al. [27], who observed that high concentrations of tea polyphenols induced bump formation in KGM-pectin films.

#### 3.1.5. Thermal Weight Analysis (TGA)

Food packaging is exposed to high temperatures and humidity during industrial production. Therefore, it was meaningful to detect the thermal stability of the sample by TGA [28]. The thermogravimetric behavior of SA-KGM-EGCG edible films was mainly divided into three stages. In stage 1 (25–200 °C), the mass of the film was lost, which was mainly due to the film-bound water evaporating [29]. The mass loss in stage 2 (200–310 °C) was related to the pyrolysis of the plasticizer glycerol, the pyrolysis of the glycosidic bond and the evaporation of the loaded EGCG [30]. In addition, since many hydrogen bonds were generated between EGCG and KGM, a large amount of heat was required to break the hydrogen bonds. The third stage (310–600 °C) was mainly the carbonization stage, which was due to the scission of the main chain of KGM and SA polymers and the overcoming of physical forces [31]. As shown in Figure 1d, films with higher EGCG content exhibited reduced overall mass loss, indicating enhanced hydrogen bonding between EGCG and SA-KGM [32]. Therefore, more energy was required to overcome the hydrogen force in EGCG and SA-KGM. It suggested that EGCG enhanced the three-dimensional network structure of SA-KGM films, thereby improving the composite films [33]. These findings align with Sun et al.’s report on EGCG’s thermal enhancement effects [34]. Although TGA showed enhanced thermal stability (Stage 2 mass loss reduction) with EGCG addition—indicating stronger internal hydrogen bonding—the simultaneous decrease in both moisture content (Figure 2b) and contact angle (Figure 2c) reveals a preferential bonding hierarchy: EGCG’s phenolic -OH first saturates SA/KGM’s binding sites (reducing bulk water absorption), then excess -OH groups increase surface hydrophilicity. This explains why 0.6% EGCG achieved an optimal balance between internal crosslinking (TGA/FTIR) and surface properties.

#### 3.1.6. Mechanical Properties

The mechanical properties of films directly influence their ability to protect food during storage. The key metrics used to assess these properties are tensile strength (TS) and elongation at break (EB). These indicators depend on the interaction between diverse components and are typically measured through tensile testing. Table 1 displays the mechanical properties of the SA films (6.94 MPa and 47.12%, respectively). The higher concentrations of EGCG showed a notable boost in tensile strength. This effect stemmed from an increased presence of reactive groups, promoting cross-linking in EGCG and SA-KGM. Thus, the enhanced cross-linking contributed to improved tensile strength in the composite films.

Polyphenolic compounds share common traits owing to their numerous phenolic hydroxyl groups. These groups have the capability to create hydrogen or covalent bonds, thereby bolstering the tensile strength index of the sample. For instance, incorporating ferulic acid into films notably augmented their tensile strength, primarily attributable to the formation of covalent bonds among the film constituents [35].

#### 3.1.7. Color and Opacity

As shown in Table 2, the incorporation of EGCG into the SA-KGM matrix decreased the L* value of samples, while the a* and b* values increased significantly. Thus, the SA-KGM-EGCG films were more vivid than the control group.

In Table 2, when EGCG was incorporated into the SA-KGM, the opacity of the film was first enhanced and then reduced. EGCG can affect the opacity of the films [36]. Specifically, some light waves may be absorbed by the pigments in EGCG, resulting in greater opacity [37]. An appropriate amount of EGCG was compatible with the films, which made the structure denser, affected the transmission of light, and increased the opacity. When the EGCG content was 0.2% (*w*/*v*), the films opacity obviously increased. However, the opacity of the films was the lowest with an EGCG content of 0.8% (*w*/*v*). This could be because the film with 0.2% (*w*/*v*) EGCG had a dense structure, which hindered the transmission of light and intensified the sample opacity. When the EGCG content rose to 0.8% (*w*/*v*), the excess EGCG may have broken the structure of SA and KGM, reducing its opacity. Furthermore, scanning electron microscope results indicated that the film’s surface with 0.2% (*w*/*v*) EGCG content was denser, and the films with 0.8% (*w*/*v*) EGCG content had a few small bumps on the surface. Similar results were found by adding lycopene microcapsules to SA-KGM films [38]. In general, the SA-KGM films with EGCG had a deeper color and higher opacity, which indicated the films could block the light effectively.

#### 3.1.8. Oxygen Permeability (OP)

Oxygen permeability (OP) serves as a gauge for evaluating the ability of films to impede oxygen transmission. Typically, a low OP is essential to mitigate food oxidation. As shown in Figure 2a, the OP of the SA/KGM films was 0.206 [cm^3^/(m^2^·24 h·0.1 MPa)]. When EGCG was incorporated into SA/KGM, the sample OP first decreased and then increased. When the content of EGCG was 0.2% (*w*/*v*), the OP was 0.08 [cm^3^/(m^2^·24 h·0.1 MPa)], indicating that EGCG increased the oxygen barrier properties of films. Generally, the OP of films was related to the microstructure, especially the void volume and the structure of the polymer chains [39]. This may be the result of EGCG filling the pores of the edible films and increasing their density, as confirmed with SEM. In addition, EGCG has a polyphenolic structure. This structure easily interacts with the hydroxyl groups in SA and KGM to augment the oxygen barrier ability of the edible films. This could prove advantageous in bolstering the preservation capabilities of the films [40]. When the EGCG content was 0.8% (*w*/*v*), the oxygen permeation rate rose significantly to 0.56 (cm^3^/(m^2^·24 h·0.1 MPa)). This may be attributed to excessive EGCG, which could make the edible films into voids and reduce the oxygen barrier property.

#### 3.1.9. Moisture Content and Contact Angle

Figure 2b reveals that the incorporation of EGCG (0–0.6%, *w*/*v*) into SA-KGM f reduced moisture content from 32.56 to 17.74. However, increasing the concentration of EGCG from 0.6 to 0.8% (*w*/*v*) had no obvious influence on the moisture level of the composite films (17.74 to 19.38%).

The affinity of a material for water was represented by assessing surface wettability. Figure 2c illustrates the contact angles observed in SA-KGM composite films with various ratios of EGCG. The contact angles decreased after adding EGCG. The prevailing factor was the hydrophilic groups present in EGCG, which facilitated the formation of hydrogen bonds. Hence, this resulted in a smaller contact angle of the samples. Similar behavior was exhibited by carboxymethyl chitosan [41].

#### 3.1.10. Antioxidant Properties

Studies have shown that phenolic compounds significantly increase the antioxidant capacity of edible films [42]. The antioxidant performance of these films was detected. Figure 2d reveals that the ABTS and DPPH radical scavenging activities of SA-KGM-EGCG films significantly increased from 55.14% to 91.47% and from 10.6% to 89.32%. This difference may be attributed to EGCG’s ability to reduce free radicals via its single electron donor properties [43]. The above findings are in line with those of previous research [44]. Therefore, the prepared films are very valuable in food storage to mitigate quality degradation stemming from oxidation. The ABTS scavenging increase from 55.14% to 91.47% (Figure 2d) directly reflected EGCG’s phenolic -OH availability, as confirmed by FTIR broadening at 3366 cm^−1^ (unbound -OH stretching). However, at 0.8% EGCG, the opacity decrease (Table 2) suggested -OH self-association (XRD crystallinity increase), which paradoxically reduced antioxidant efficacy despite higher EGCG loading.

### 3.2. Application in Mandarin Fish Preservation

#### 3.2.1. pH Value

As displayed in Figure 3a, the pH value of all fish samples decreased first and then increased. The decline in pH value may be due to the formation of acids from the breakdown of muscle adenosine triphosphate (ATP) and glycogen glycolysis, which would decrease the pH. The main reason for the increase in pH was the destruction of muscle tissue by microorganisms, leading to the accumulation of alkaline substances such as amine compounds. This suggests that the fresh meat began to decompose and deteriorate [45]. The pH of the control and SA-KGM groups began to rise on the third day, indicating the onset of decomposition, while the SA-KGM-EGCG group exhibited an elevated pH on the sixth day, indicating that EGCG can effectively delay fish spoilage. In fact, EGCG exhibits antibacterial and antioxidant properties, effectively inhibiting the proliferation of spoilage microorganisms, retarding enzyme activity, and reducing the accumulation of alkaline substances [46].

#### 3.2.2. TBARS

TBARS values measure the level of fat oxidative rancidity in aquatic products and meat. The difference in TBARS values of the fish pieces wrapped with nothing (control), SA-KGM film, and the SA-KGM-EGCG films can be seen in Figure 3b. The initial TBARS value of fish pieces was assumed to be consistent across all cases before storage. An increase in TBARS values was due to the oxidation of fish fat into simpler compounds like peroxides [47]. After 3 days of storage, the TBARS values of control, SA-KGM, and SA-KGM-EGCG groups were 0.78, 0.69, and 0.59 mg MDA/100 kg meat. This result demonstrated the rapidly activated protection of SA-KGM films against fat oxidation in fish. However, the influence of the SA-KGM films did not endure throughout the following storage process. As in Figure 3b, the TBARS of the SA-KGM group obviously respectively increased to 0.91, 1.57, and 1.91 mg MDA/100 kg meat after 6, 9, and 12 days of storage. On the other hand, those of the SA-KGM-EGCG group were lower, at 0.77, 1.25, and 1.58 mg after 6, 9, and 12 days of storage. Hence, the SA-KGM films were most effective against fat oxidation in the fish pieces during the initial period of storage. Nevertheless, during the preservation period of up to 12 days, EGCG in the composite film exhibited a robust antioxidant effect.

#### 3.2.3. TVB-N

The impact of various storage durations on the TVB-N content of the fish pieces is outlined in Figure 3c. Following 3 days of preservation, the TVB-N contents of the control, SA-KGM, and SA-KGM-EGCG groups were 3.17, 2.99 and 2.70 mg/100 g meat. This result suggested that SA-KGM films can maintain the freshness of chilled fish pieces for the first 3 days. Nevertheless, the TVB-N content of the fish pieces enveloped in SA-KGM films increased significantly to 12.97, 14.74, and 22.86 mg/100 g after being stored for 6, 9, and 12 days. The TVB-N of fish pieces wrapped with the SA-KGM-EGCG films rose from 9.52 to 12.20 to 22.02 mg/100 g after being stored for 6, 9, and 12 days. The lowest TVB-N content appeared in the SA-KGM-EGCG group during the first 9 days. The rise in TVB-N content was evident, with significant differences observed between the two types of improved films. The antioxidant capacity of EGCG is credited with preserving and maintaining the freshness of fish. According to findings by Bekhit, Holman, Giteru, and Hopkins [48], fish quality meets acceptable standards when TVB-N content is below 15 mg/100 g. Hence, the SA-KGM-EGCG films exhibited superior freshness-preserving properties compared to other films of the same type and could effectively preserve fish for up to 9 days.

## 4. Conclusions

This study successfully established a protocol for fabricating functional polysaccharide-based edible films and demonstrated their efficacy in aquatic product preservation. The results indicate that SA-KGM-EGCG composite films with 0.6% EGCG concentration exhibited optimal performance. Through hydrogen bond crosslinking (evidenced by FTIR shifts of -COO- groups), the films formed homogeneous amorphous networks (XRD peak broadening at 21°), achieving enhanced tensile strength (9.21 MPa) and significantly improved oxygen barrier properties (OP value reduced to 0.08 cm³/m^2^·24 h·0.1 MPa). Simultaneously, these films demonstrated peak antioxidant activity (plateau in ABTS radical scavenging capacity) attributed to phenolic -OH saturation (FTIR peak broadening at 3366 cm^−1^). In mandarin fish preservation trials, the 0.6% EGCG composite film effectively maintained product quality, keeping low TVB-N and TBARS values for 9 days. These findings provide critical insights for developing high-performance edible packaging materials with tailored functionalities.

## Figures and Tables

**Figure 1 foods-14-01570-f001:**
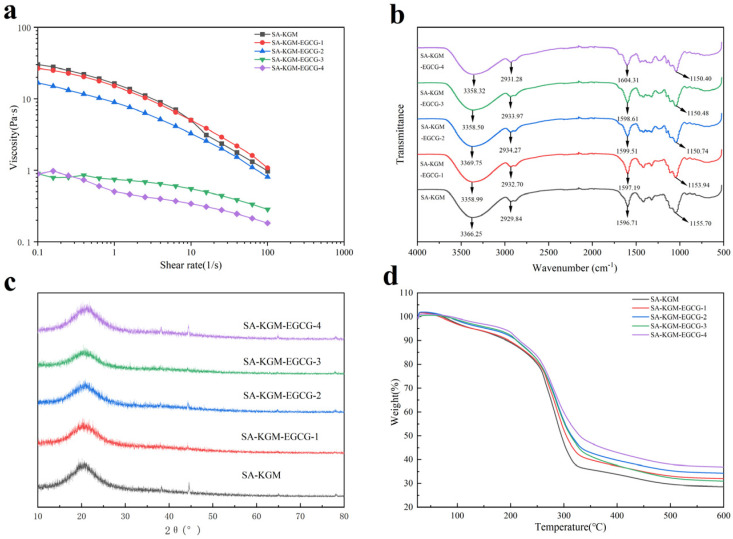
(**a**) Rheological properties of film-forming fluid, (**b**) FTIR spectra of edible films, (**c**) XRD spectra of edible films (**d**) TGA spectra of edible films.

**Figure 2 foods-14-01570-f002:**
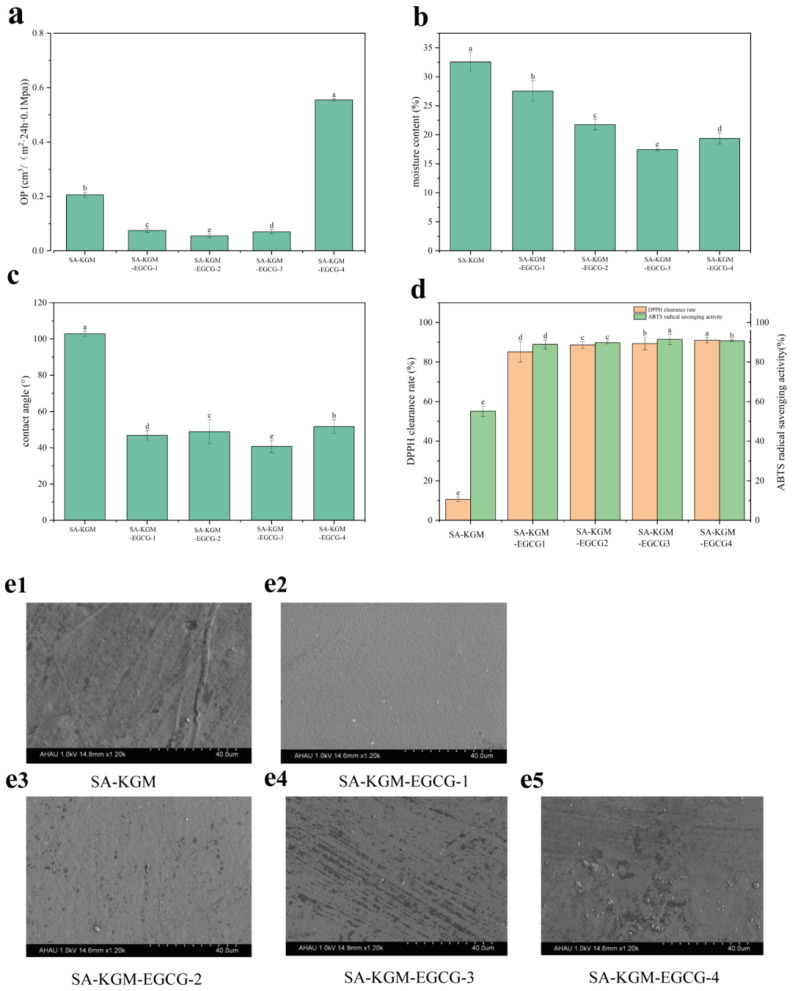
(**a**) Opacity of antioxidant edible films, (**b**) moisture content of edible films, (**c**) contact angle of edible films, (**d**) antioxidant properties of antioxidant edible films, (**e1**–**e5**) SEM micrographs of the cross-sectional images of composite films with different concentrations of EGCG. Data marked with different superscript letters in the figure indicate statistically significant differences (*p* < 0.05).

**Figure 3 foods-14-01570-f003:**
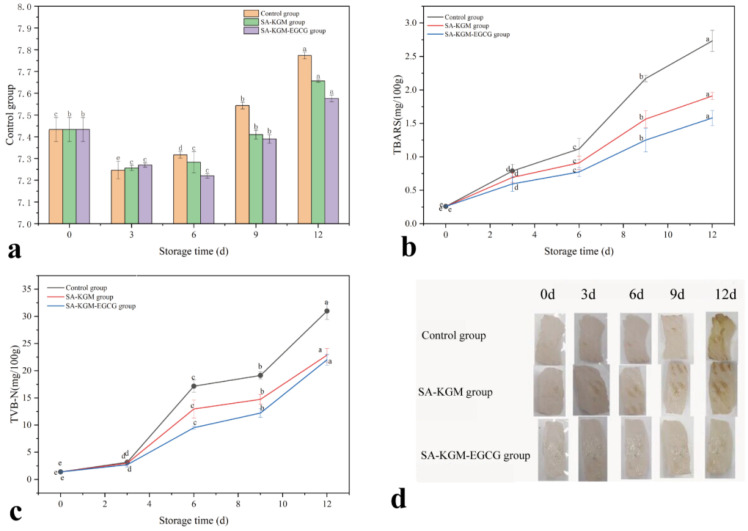
(**a**) Changes in pH values of mandarin fish fillets stored at 4 °C during 12 days, (**b**) changes in TBARS values of mandarin fish fillets stored at 4 °C during 12 days, (**c**) changes in TVB-N values of mandarin fish fillets stored at 4 °C during 12 days, (**d**) The appearance of mandarin fish slices under three different films during 12 days. Data marked with different superscript letters in the figure indicate statistically significant differences (*p* < 0.05).

**Table 1 foods-14-01570-t001:** Thickness and mechanical properties of edible film.

Film Samples	Thickness (mm)	TS (MPa)	EAB (%)
SA-KGM	0.16 ± 0.04 ^b^	6.94 ± 1.33 ^b^	47.12 ± 3.07 ^a^
SA-KGM-EGCG1	0.15 ± 0.01 ^c^	9.44 ± 2.06 ^b^	39.21 ± 2.61 ^a^
SA-KGM-EGCG2	0.18 ± 0.01 ^a^	12.34 ± 1.44 ^ab^	38.46 ± 0.08 ^a^
SA-KGM-EGCG3	0.18 ± 0.01 ^a^	12.68 ± 5.12 ^ab^	28.08 ± 5.46 ^b^
SA-KGM-EGCG4	0.15 ± 0.01 ^c^	17.68 ± 5.62 ^a^	26.25 ± 10.45 ^b^

All values are presented as mean ± standard deviation. Data in the same line with different superscript letters are significantly different (*p* < 0.05). TS = Tensile strength (MPa); EAB = Elongation at break (%).

**Table 2 foods-14-01570-t002:** Color and opacity properties of edible film.

Film Samples	L*	a*	b*	Opacity
SA-KGM	72.67 ± 1.21 ^a^	0.68 ± 0.01 ^e^	−1.96 ± 0.09 ^e^	1.12 ± 0.13 ^d^
SA-KGM-EGCG1	63.57 ± 1.57 ^c^	2.20 ± 0.01 ^a^	14.88 ± 0.36 ^a^	2.60 ± 0.45 ^a^
SA-KGM-EGCG2	64.18 ± 1.19 ^bc^	2.17 ± 0.19 ^b^	11.16 ± 0.73 ^b^	1.96 ± 0.15 ^b^
SA-KGM-EGCG3	65.59 ± 1.92 ^b^	1.71 ± 0.12 ^c^	7.32 ± 1.00 ^c^	1.68 ± 0.13 ^c^
SA-KGM-EGCG4	64.69 ± 1.83 ^b^	1.59 ± 0.28 ^d^	6.55 ± 0.81 ^d^	1.09 ± 0.10 ^e^

All values are presented as mean ± standard deviation. Data in the same line with different superscript letters are significantly different (*p* < 0.05). L* = Lightness; a* = Redness/greenness; b* = Yellowness/blueness.

## Data Availability

The original contributions presented in this study are included in the article. Further inquiries can be directed to the corresponding author.

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
