# Peer review of "Preparation of Epigallocatechin Gallate-Enriched Antioxidant Edible Films Based on Konjac Glucomannan and Sodium Alginate: Impact on Storage Stability of Mandarin Fish"

_foods, 2025, doi:10.3390/foods14091570_

Round 1
Reviewer 1 Report
Comments and Suggestions for Authors
Manuscript ID: foods-3483691
Title: Preparation of epigallocatechin gallate-enriched antioxidant edible films based on konjac glucomannan and sodium alginate: impact on storage stability of mandarin fish.
General comments:
The manuscript aimed to develop an antioxidant edible film based on konjac glucomannan (KGM), sodium alginate (SA), and epigallocatechin gallate (EGCG) for preserving mandarin fish. The study explores the structural, mechanical, and antioxidant properties of the films, aiming to enhance food storage stability while reducing reliance on synthetic preservatives. If validated with stronger experimental design, the findings could contribute to sustainable food packaging and natural preservation techniques, aligning with the growing demand for biodegradable and functional food packaging materials. However, the manuscript does not deliver important innovations that would enhance the understanding established by previous research.
- The study focuses on konjac glucomannan (KGM) and sodium alginate (SA) edible films incorporating epigallocatechin gallate (EGCG), which is not a novel concept. Numerous studies have already explored polysaccharide-based films enriched with polyphenols for food preservation. The manuscript does not sufficiently differentiate itself from previous research (e.g., no clear advancements in film formulation or functional improvements beyond what is already published).
- The film preparation method lacks clarity regarding critical parameters such as film uniformity and reproducibility. The film drying conditions (50°C for 18h) may affect EGCG stability, but this is not investigated.
- While antioxidant and mechanical properties are discussed, other crucial functional properties, such as biodegradability, antimicrobial efficacy against foodborne pathogens, and real-world application in different food matrices, are not assessed.
- The storage test for mandarin fish is limited to 12 days at 4°C. There is no comparison with commercial food packaging materials to justify the superiority of the proposed films. Moreover, sensory evaluation is missing, which is essential in food applications.
- The manuscript presents various experimental results, but statistical rigor is lacking. Many results, particularly in film characterization (SEM, XRD, FTIR), are discussed qualitatively without quantitative validation.
- The TVB-N and TBARS data show some reduction in spoilage indicators, but the observed differences between the control and treatment groups are relatively small, and it is unclear if they are practically significant.
- EGCG is known to be highly sensitive to oxidation and pH variations, which could lead to degradation. The manuscript does not analyze whether EGCG remains stable throughout storage.
- The XRD and FTIR data suggest interactions between EGCG and the film matrix, but without further molecular characterization (e.g., DSC or detailed rheological analysis), it is unclear if these interactions affect EGCG’s bioactivity over time.
- Improve Figures image quality, particularly SEM and fish appearance figures.
- Include baseline values for fish storage tests (e.g., initial pH, TBARS).
- Provide quantitative comparisons with existing literature to justify findings.
- The manuscript contains numerous grammatical errors and awkward phrasing, which hinder readability.
- Several key claims lack proper citations, and some references appear outdated or do not directly support the conclusions drawn.
Finally, the manuscript attempts to develop and characterize antioxidant edible films based on konjac glucomannan (KGM), sodium alginate (SA), and epigallocatechin gallate (EGCG) for the preservation of mandarin fish. While the topic is relevant to food packaging and preservation, the study lacks significant novelty, as similar polysaccharide-based films with polyphenol incorporation have been widely studied. Additionally, the manuscript suffers from methodological weaknesses, insufficient statistical analysis, and unclear data interpretation. Key aspects such as EGCG stability, biodegradability, and antimicrobial efficacy are not adequately addressed. Given these limitations, the study does not provide a substantial contribution to the field and requires major revisions before being considered for publication.
Comments on the Quality of English LanguageNo comments
Reviewer 2 Report
Comments and Suggestions for Authors
Please find the attachment

Dear Editor
The manuscript is written casually. The manuscript has scope for improvement
Round 2
Reviewer 1 Report
Comments and Suggestions for Authors
After reviewing the changes and suggestions made, I find the article suitable for publication.
However, I recommend that all authors carefully re-read the document and improve its language style.
I recommend that all authors carefully re-read the document, improve its language style, and fix all typos.
Author Response
Comments 1:I recommend that all authors carefully re-read the document, improve its language style, and fix all typos
Response 1:
Thank you for pointing this out. We agree with this comment. We will carefully review the language used in the document to ensure it meets standard accuracy requirements.
Reviewer 2 Report
Comments and Suggestions for Authors
Dear Author
I recommend you to reduce the plaigarism to less than 15% which is 19% now
Author Response
Comments 1:
I recommend you to reduce the plaigarism to less than 15% which is 19% now
Response 1:
Thank you for pointing this out. We agree with this comment. Therefore, I will follow your requirements to reduce the plagiarism rate.